# Identification of *TIAM1* as a Potential Synthetic-Lethal-like Gene in a Defined Subset of Hepatocellular Carcinoma

**DOI:** 10.3390/ijms24076387

**Published:** 2023-03-28

**Authors:** Chalermsin Permtermsin, H Lalchungnunga, Sirintra Nakjang, John Casement, Laura Frances Ogle, Helen L. Reeves, Gordon Strathdee, Ruchi Shukla

**Affiliations:** 1Biosciences Institute, Newcastle University Centre for Cancer, Newcastle University, Newcastle NE2 4HH, UK; 2Bioinformatics Support Unit, Newcastle University, Newcastle NE2 4HH, UK; 3Translational and Clinical Research Institute, Newcastle University Centre for Cancer, Newcastle University, Newcastle NE2 4HH, UK; 4Liver Unit, Freeman Hospital, Newcastle-Upon-Tyne Hospitals NHS Foundation Trust, Newcastle-upon-Tyne NE7 7DN, UK; 5Department of Applied Sciences, Northumbria University, Newcastle NE1 8ST, UK

**Keywords:** synthetic lethal gene, DNA methylation, HCC, TIAM1, RAC1

## Abstract

Hepatocellular carcinoma (HCC), the most common type of liver cancer, has very poor outcomes. Current therapies often have low efficacy and significant toxicities. Thus, there is a critical need for the development of novel therapeutic approaches for HCC. We have developed a novel bioinformatics pipeline, which integrates genome-wide DNA methylation and gene expression data, to identify genes required for the survival of specific molecular cancer subgroups but not normal cells. Targeting these genes may induce cancer-specific “synthetic lethality”. Initially, five potential HCC molecular subgroups were identified based on global DNA methylation patterns. Subgroup-2 exhibited the most unique methylation profile and two candidate subtype-specific vulnerability or SL-like genes were identified for this subgroup, including TIAM1, a guanine nucleotide exchange factor encoding gene known to activate Rac1 signalling. siRNA targeting TIAM1 inhibited cell proliferation in TIAM1-positive (subgroup-2) HCC cell lines but had no effect on the normal hepatocyte HHL5 cell line. Furthermore, *TIAM1*-positive/subgroup-2 cell lines were significantly more sensitive to the TIAM1/RAC1 inhibitor NSC23766 compared with *TIAM1*-negative HCC lines or the normal HHL5 cell line. The results are consistent with a synthetic lethal role for *TIAM1* in a methylation-defined HCC subgroup and suggest it may be a viable therapeutic target in this subset of HCC patients.

## 1. Introduction

Liver cancer was the seventh most common type of cancer and the second leading cause of cancer death worldwide in 2018, according to the GLOBOCAN estimates [1]. Hepatocellular carcinoma (HCC) is the most prominent type of liver cancer with poor 5-year survival [2]. The poor prognosis is associated with late patient diagnosis as the disease usually develops on a background of chronic liver disease (CLD) and has no distinct or specific symptoms in its early stages. CLD can develop either due to viral infections, such as hepatitis B and C virus, excessive alcohol intake, exposure to toxins, or metabolic syndrome, due to obesity and type 2 diabetes. Although risk factors such as HBV and HCV are now under control and declining due to the development of successful vaccines and treatments, respectively, the obesity and diabetes epidemics are growing. Hence, the incidence and mortality from HCC are predicted to rise more rapidly than any other cancer [3]. Coexistent CLD limits the use of cytotoxic or medical therapies. Available palliative treatments (for example sorafenib, lenvatinib, regorafenib) are poorly tolerated and the expected median survival is approximately 12 months [4]. Immuno-oncology (I-O) therapies potentially offer new hope for patients with HCC. In clinical trials, over 2000 HCC patients have now been treated with checkpoint inhibitors (CKIs) targeting PD-1/PDL1. The treatments are well tolerated and anti-tumour responses are evident. However, only about 15% of patients respond to monotherapy [5]. Up to 27% respond to the CKI atezolizumab combined with the vascular endothelial growth factor (VEGF) inhibitor bevacizumab [6], but the side effect profile limits its use to carefully selected patients. Consequently, improving therapy for HCC patients is one of the key priorities in cancer research and there is an urgent need for the development of treatment strategies that are more effective at targeting HCC cells and exhibit greater specificity to reduce patient toxicities.

The identification of cancer-specific synthetic lethal (SL) genes is among the most promising approaches to achieving the increased success of cancer treatment whilst also reducing the burden of off-target toxicities [7]. The SL genes are only required for cell survival in the presence of additional other genetic changes and if SL-partner genes can be identified for cancer-driving genetic changes, these could allow for the highly specific targeting of cancer cells without causing normal tissue toxicity [8]. The recent development of PARP inhibitors as a treatment for BRCA1/2-deficient ovarian cancer represents proof-of-principle that targeting synthetic lethal genes can improve cancer treatment [9].

We have recently developed a novel bioinformatic approach and demonstrated that it has the ability to identify cancer-subtype-specific SL genes, also referred to as subtype-specific vulnerability (SSV) genes, with a very high degree of specificity [10]. This approach is based on comparative analysis of different subtypes within a single type of cancer and allows for the identification of specific genes that have selectively failed to acquire methylation in one specific subtype. Functional analysis of the candidates identified across two separate cancer types (acute lymphoblastic leukaemia and medulloblastoma) demonstrated confirmation of the identified candidates as functional synthetic lethal genes in the majority (77%) of identified candidate genes. As the cancer-specific disruption of genome-wide DNA methylation patterns that underlie this approach are broadly similar in all cancer types [11], this approach should be widely applicable for the identification of novel synthetic lethal targets across most cancer types, including HCC.

Thus, in this study we have used genome-wide DNA methylation patterns to identify different molecular subgroups of HCC (as the identification of multiple molecular subgroups is a requirement for the subsequent SSV gene identification). Focusing on the most unique of the identified subgroups (referred to as subgroup-2), we identified two candidate synthetic-lethal-like/SSV genes, including *TIAM1*, a known regulator of RAC1 signalling. Functional assessment of the role of *TIAM1* in cell line models confirmed that it exhibited properties consistent with being an HCC subgroup-2-specific SSV gene.

## 2. Results

### 2.1. Identification of Potential Molecular Subgroups in HCC

The bioinformatics approach that we have previously developed for the identification of SSV genes requires the separation of samples into multiple genetic or epigenetic subtypes [10]. To this end, we used non-negative matrix factorization (NMF) [12] to allow for the clustering of HCC patients into DNA-methylation-based subgroups. The analysis was performed using a publicly available data set of 221 HCC samples that had been collected as part of the HEPTROMIC Consortium and for which genome-wide DNA methylation data was available [13]. This analysis found that the most stable solution involved five potential subgroups (hereafter referred to as subgroups 1–5) (Figure 1). However, as can be seen from Figure 1, the extent to which the subgroups exhibit clearly differential regions of methylation is limited. Of the five subgroups, only subgroup-2 exhibits clearly unique regions of differential methylation (Figure 1). From our previous analysis of other cancer types [10], SL-like genes identified by this approach are typically identified as having methylation patterns similar to normal cells and, consistent with this, the regions in subgroup-2 that are differential methylated in comparison to the other four subgroups, typically strongly resemble normal liver tissue. Thus, the subsequent SSV gene analysis focused specifically on this subgroup. In contrast to the clustering seen for the methylation data, using whole transcriptome data, these subgroups do not exhibit any clear differential clustering (Appendix A).

### 2.2. Identification of Subgroup-2-Specific SSV Gene Candidates

Subsequently, we applied the novel bioinformatics approach which we have developed [10] to identify potential SL-like/SSV genes specific for methylation subgroup-2. This approach allows the identification of SSV genes through integrative analysis of genome-wide DNA methylation and gene expression data. Focusing specifically on the most differentially methylated cluster (subgroup-2), the analysis identified two candidate SSV genes; lactate dehydrogenase B (*LDHB*) (Figure 2A) and T-cell lymphoma invasion and metastasis 1 (*TIAM1*) (Figure 2B). For both candidates, samples mapping to subgroup-2 exhibit pronounced lower methylation levels, which were also associated with increased expression (Figure 2). Our previous work on other cancer types has shown that subgroup-specific methylation/expression patterns of this type are frequently associated with subgroup-specific SL-like genes [10].

### 2.3. Identification of Cell Line Models Based on Methylation/Expression of the Identified SSV Gene Candidates

Multiple HCC cell lines were assessed to identify lines that could be used as models for HCC subgroup-2 (to allow testing of the functional relevance of *TIAM1*), as well as controls that could be used as models for non-HCC-subgroup-2 (i.e., derived from HCC not associated with subgroup-2). Based on the primary sample HCC data, subgroup-2 lines should exhibit high expression and low methylation of both *TIAM1* and *LDHB*, while non-HCC-subgroup-2 lines would exhibit low expression and high methylation for both genes. As shown in Figure 3, two of the four tested HCC cell lines (PLC/PRF-5 and SNU-182) exhibited high expression of both genes, while the other two cell lines (HepG2 and Huh-7) expressed very low levels of both genes. The normal, immortalised, human hepatocyte cell line HHL5 was also found to express similar levels of *TIAM1* and *LDHB* as seen in the proposed subgroup-2-specific cell lines. Although PLC/PRF-5 exhibited *LDHB* expression significantly lower than HHL5 (*p* < 0.0001, one-way ANOVA with multiple comparisons using HHL5 as the base sample), it was significantly higher than HepG2 and Huh-7 (*p* < 0.0001, one-way ANOVA with multiple comparisons using PLC/PRF-5 as the base sample). This is in line with the SSV gene hypothesis that the genes are expressed in normal cells but are then inactivated in transformed cells except for the cancer type where cells are dependent on the gene.

In addition, analysis of DNA methylation at both loci found the predicted corresponding low levels of methylation in the proposed subgroup-2 cell lines and high methylation in the non-subgroup-2 lines (Figure 4). Of particular note was, for both genes, the region of differential methylation between the proposed subgroup-2 (PLC/PRF-5, SNU-182) and non-subgroup-2 (HepG2, Huh-7) cell lines corresponded exactly to the region of differential methylation identified in the HCC subgroup-2 primary samples, thus providing good evidence that both PLC/PRF-5 and SNU-182 are likely to be derived from HCC subgroup-2 primary samples (Figure 4). Our bioinformatics approach for the identification of subgroup-specific SSV genes is dependent on identifying selective retention of normal DNA methylation levels in a single molecular subgroup and thus the methylation patterns at the identified SSV genes usually mirror those seen in corresponding normal cells. Consistent with this hypothesis, the methylation pattern across *TIAM1* in the subgroup-2 HCC cell lines was highly correlated with methylation in the normal HHL5 cell line (r^2^ = 0.75, *p* = 0.0005) and methylation levels across the differential methylated region (DMR) were very similar (average methylation across the *TIAM1* DMR was 16% vs. 15% vs. 62% in HHL5, subgroup-2 HCC cell lines, and non-subgroup-2 HCC cell lines, respectively, Appendix A). For *LDHB*, there was only a partial retention of normal methylation levels in the HCC subgroup-2 cell lines (average methylation across the *LDHB* DMR was 4% vs. 43% vs. 81% in HHL5, subgroup-2 HCC cell lines, and non-subgroup-2 HCC cell lines, respectively).

### 2.4. siRNA-Mediated Knockdown of TIAM1 Inhibits Growth in HCC Subgroup-2-like Cell Line Models

If *TIAM1* is a genuine SSV gene in HCC subgroup-2, then inhibiting its expression should be able to induce specific killing of HCC subgroup-2 cells. To assess this possibility, the identified subgroup-2 cell lines were transfected with *TIAM1*-specific or non-silencing siRNAs. As a control, the normal hepatocyte HHL5 cell line, which also expresses TIAM1, was also included. *TIAM1*-specific siRNA efficiently reduced the transcript level of *TIAM1* in all the three cell lines (Figure 5A). However, assessment of cell growth following *TIAM1* knockdown identified significantly reduced growth in both HCC subgroup-2 cell lines, but not in the normal HHL5 cell line (Figure 5B).

### 2.5. HCC Subgroup-2-like Cell Line Models Show Increased Sensitivity to the TIAM1/RAC1 Inhibitor NSC23766

To further confirm if *TIAM1* exhibited SSV-gene-like functional properties, we assessed the impact of the treatment of HCC cell lines and normal HHL5 cells with the inhibitor NSC23766. TIAM1 functions as an activator of RAC1 and NSC23766 inhibits the TIAM1/RAC1 interaction [14]. Cell growth was followed for 6 days post-treatment and the IC50 dose for each cell line (at day 6) was calculated. As can be seen from Table 1, the HCC subgroup-2 cell lines exhibited very similar and significantly lower IC50 values compared with the non-HCC-subgroup-2 cell lines and the normal HHL5 cell line (even though this cell line expresses *TIAM1*) (*p* = 0.007) (also see Appendix A). Consistent with the results obtained by siRNA knockdown, these results suggest that *TIAM1* is an essential gene for the optimal growth of HCC subgroup-2 cells, but is not an essential gene in other HCC subgroups or in normal hepatocytes. The impact of NSC23766 may primarily be related to the inhibition of proliferation, as IC50 doses of NSC23766 were not associated with the induction of cell death in any of the cell lines (Appendix A). There was limited evidence of subgroup-2-specific cell death at higher doses (IC50 × 4) but this was restricted to the SNU182 cell line (Appendix A).

## 3. Discussion

In this study, we have investigated the potential for the identification of HCC-specific synthetic lethal genes that could represent novel therapeutic targets in HCC. This led to the identification of *TIAM1* as a potential synthetic-lethal-like SSV gene in a subset of HCC patients. Initial assessment of the effect of genetic or drug-based targeting of *TIAM1* supported its specific requirement in the subset of HCC cells that retain *TIAM1* expression and suggest that *TIAM1* merits investigation as a potential therapeutic target in a subset of HCC.

This identification of potential synthetic lethal genes was based on a recently developed bioinformatics approach that requires knowledge of defined molecular subtypes within the cancer being studied. The initial stage of the study used NMF to define different subgroups based on genome-wide DNA methylation data. The most stable solution from this approach defined five HCC subgroups. However, the extent of differential methylation between the subgroups was limited. This is in contrast to other cancers analysed in a similar fashion, such as in neuroblastoma [15] or medulloblastoma [16]. Consequently, the subsequent SSV gene analysis was limited to the one subgroup (subgroup-2) that displayed more extensive regions of subgroup-specific differential methylation. Moreover, unique DMRs in subgroup-2 looked similar to normal, non-cancer controls (normal hepatocytes and cirrhotic livers). This is consistent with our observations in other cancer types, where SSV genes are characterized by the absence of background accumulation of methylation seen in subtypes where the gene is not required for cell survival [10]. Thus, additional approaches to derive molecular subgroups representative of all HCC may be required to allow analysis of the remaining HCC cases. It is possible that the lack of clear subgrouping may relate to a comparatively low tumour cell content in the HCC samples. Such issues could potentially be addressed through the use of cohorts in which tumour cells are specifically purified or through bioinformatic deconvolution approaches to allow for the delineation of methylation profiles that are tumour cell specific. Alternatively, HCC may lack clearly distinct molecular subtypes and more disease-wide analysis approaches may be required.

TIAM1 is a guanine nucleotide exchange factor, which has previously been reported to function as a specific activator of RAC1, which does not impact on the activity of RAC-1-like proteins CDC42 or RhoA signalling [14]. RAC1 has been identified as playing important roles in cancer, particularly in the control of cell motility and metastasis [17], and has also been implicated in cancer cell proliferation [18]. Thus, retention of TIAM1 expression in HCC subgroup-2 may be crucial for this subset of HCC cases due to activation of RAC1 signalling. Consistent with this, we found that subgroup-2 cell line models were preferentially sensitive to the RAC1 inhibitor NSC23766. Furthermore, although the normal HHL5 cell line expresses TIAM1, it showed no evidence of increased sensitivity compared with TIAM1-negative non-subgroup-2 HCC lines, consistent with the increased sensitivity being specific for transformed and not normal hepatocytes. However, the impact of drug treatment was not specific for subgroup-2 cell lines and sensitivity at higher concentrations of NSC23766 was still observed in the TIAM1-negative HCC cell lines. This indicates that NSC23766 either exhibits RAC1-independent off-target effects or that RAC1 activation due to alternative upstream regulators also plays a role in non-subgroup-2 HCC and in normal hepatocytes. Thus, it remains to be elucidated the extent to which retention of *TIAM1* expression in HCC subgroup-2 suggests a general dependence on RAC1 signalling or a specific dependence on RAC1 activation by TIAM1. Recent evidence has demonstrated that RAC1-activating proteins such as TIAM1 also have roles in modulating the downstream impact of RAC1 activation, resulting in GEF-specific signalling cascades [19]. As NSC23766 is known to inhibit RAC1 activation by other GEFs in additional to TIAM1 [14], it may be necessary to develop more TIAM1-specific inhibitors to enable targeting of subgroup-2 HCC.

siRNA-mediated knockdown of TIAM1 was able to significantly inhibit cell growth in HCC subgroup-2 cell lines, but not in the immortalised normal hepatocyte cell line HHL5, even though it too expresses TIAM1. This result was consistent with the preferential sensitivity to NSC23766. A limitation of siRNA-mediated gene knockdown is that the knockdown is partial and transient. This could be overcome by approaches such as CRISPR, which can allow for the derivation of complete and stable gene inactivation. However, selection for such stable inactivation would not be possible for functional synthetic lethal genes. However, inducible approaches, such as the recently reported Cre-Controlled CRISPR mutagenesis [20], could be employed to more clearly determine if loss of *TIAM1* expression is specifically toxic to HCC subgroup-2 cells.

Overall, this study demonstrates a potential synthetic lethal function for *TIAM1* in a subset of HCC patients defined by differential genome-wide DNA methylation. Based on this initial observation, further work will be required to determine if strategies for targeting TIAM1 could be developed as potential new tumour-specific therapeutic approaches for a defined subset of HCC patients.

## 4. Materials and Methods

### 4.1. Cell Culture

Hepatocellular carcinoma (HCC) cell lines HepG2, Huh-7, and PLC/PRF-5 were cultured with Dulbecco’s modified Eagle’s/Ham F12 medium (Sigma, Dorset, UK). Hepatocellular carcinoma SNU182 cell line was cultured with RPMI-1640 and the immortalised hepatocyte cell line HHL5, was propagated in Dulbecco’s modified Eagle’s medium/high glucose supplemented with 1% non-essential amino acid (Sigma, UK). All mediums were supplemented with 10% fetal bovine serum (Gibco, Loughborough, UK), 100 U/mL penicillin and 100 µg/mL streptomycin (Sigma), and 1% L-glutamine (Sigma). Cells were incubated at 37 °C in a humidified incubator with 5% CO_2_. All cell lines were authenticated (Northgene, Deeside, UK) and regularly screened for mycoplasma contamination.

#### Assessment of Sensitivity to RAC1 Inhibitor NSC23766

HepG2, Huh-7, SNU182, PLC/PRF-5, and HHL5 cell lines were seeded at 2000 cells/well into 96-well plates. After incubating overnight at 37 °C, 5% CO_2_, NSC23766 was added at a concentration of 0, 25, 50, 75, and 100 µM in a final volume of 200 µL and cells were incubated for a further six days. Cell viability was assessed using the MTT assay. In short, 20 μL of MTT solution (5 mg/mL in PBS) (Abcam, Cambridge, UK) was added to each well, and the cells were incubated at 37 °C for 4 h. Medium was discarded and replaced with 150 µL DMSO into each well to dissolve the formazan crystals. The optical density was measured at a wavelength of 570 nm, using a microplate reader (FLUOstar Omega, BMG Labtech, Aylesbury Bucks, UK). DMSO was used as the blank. Results are presented as averages of three independent replicates.

### 4.2. siRNA Transfection of Mammalian Cells

*TIAM1*-specific siRNA was used to knockdown the *TIAM1* gene expression in cell lines. All cell lines were seeded at 8 × 10^4^ cells/well into 12-well plates to achieve about 70% confluency after overnight incubation. siRNA (either TIAM1-specific (sc-36669, Santa Cruz Biotechnology, TX, USA) or non-targeting control (AllStars Neg. Control siRNA, Qiagen, Manchester, UK), final concentration of 50 nM) were transfected using *Tran*IT-X2 transfection reagent (Mirus, WI, USA) as per the manufacturer’s instructions. The transfected cells were cultured for 48 h before further analysis via RT-qPCR to confirm the knockdown. Cell proliferation was determined with the MTT assay, as above 6 days after transfection and presented as the average of five replicates.

### 4.3. RNA Extraction and qRT-PCR

RNA extraction was performed using total RNA purification kit (Norgenbiotek, Thorold, ON, Canada) and cDNA synthesis was achieved using the high-capacity cDNA reverse transcription kit (Applied Biosystems, Loughborough, UK), both according to the manufacturer’s protocol. qRT-PCR was performed using Syber green mix (Invitrogen, Loughborough, UK) as per the instructions at 60 °C annealing temperature. PCR was run on a QuantStudio 7 Flex (Applied Biosystems). Primers used were as follows: TIAM1 (NM_003253.3) Forward–*GAAGGACTTTGTCTTCTGCC*, Reverse–*ATGGCGGTGATCCAGTTTTC* (96 bp product); LDHB (NM_002300.8) Forward–*GAAGAAGAGGCAACAGTTCC*, Reverse–*GCCACAATTTTAGGTGTCTGA* (200 bp product); HPRT1 (NM_000194.3) Forward–*TTGCTTTCCTTGGTCAGGCA*, Reverse–*ATCCAACACTTCGTGGGGTC* (85 bp product). The primers were designed using NCBI primer blast online tool (https://www.ncbi.nlm.nih.gov/tools/primer-blast/ accessed on 1 November 2017) and the analysis was done according to MIQUE-guidelines (Appendix A).

### 4.4. Genome-Wide DNA Methylation Analysis in HCC Cell Lines

Bisulfite conversion was performed using the Zymo EZ-96 DNA methylation kit (Zymo Research, Cambridge, UK). Genome-wide DNA methylation was quantified in all samples using the Infinium MethylationEPIC BeadChip microarray, which evaluates genome-wide CpG methylation at over 850,000 sites and was carried out at the Wellcome Trust Clinical Research Facility, University of Edinburgh (Edinburgh, UK). Raw methylation data from MethylationEPIC arrays for all test and control samples were processed using the minfi Bioconductor package version 1.28.4 in R studio version 3.5.3. The single-sample (ssNoob) method was used for normalization [21]. Probes with a detection *p*-value > 0.01 and cross-reactive probes (i.e., probes which cross-hybridize between autosomes and sex chromosomes) [22] were removed. After processing, 820,139 probes remained for the childhood paired samples (*n* = 820,134 of these passed quality control in the adult samples). Methylation values were transformed to β values, which ranged from 0 (0% methylation) to 1 (100% methylation), representing methylation intensity [23].

### 4.5. Identification of Molecular Subgroups Using Non-Negative Matrix Factorisation

We used the DNA methylation profile of 224 HCC samples from GSE56588 as a model dataset to determine potential HCC subgroups. Pre-processed methylation beta values were downloaded from the GEO database (accession GSE56588). Poorly performing probes and potentially confounding loci were removed as previously described [24]. The remaining probe beta values (429,627) were converted to M-scores [25] and 10,000 of the most variable probes by standard deviation were used for subgroup identification. A consensus bootstrapped non-negative matrix factorisation (NMF) clustering was performed on the filtered methylation M-scores matrix to determine the most robust subgroups in the model HCC dataset. In brief, we performed NMF and K-means clustering, testing all combination of 2–10 metagenes and clusters/subgroups with the bootstrapped resampling method (*n* = 250) of 80% of the samples at a time to test for reproducibility. Average sample subgrouping reproducibility was assessed to determine optimal combinations of metagenes and clusters.

### 4.6. Identification of Subtype-Specific Vulnerability or Synthetic-Lethal-like Gene Candidates in HCC

Differential methylation analysis in primary HCC was performed using a publicly available data set of genome-wide DNA methylation for 221 HCC samples that had been collected as part of the HEPTROMIC Consortium [13], using Infinium HumanMethylation450 arrays (Illumina, CA, USA). Differentially methylated regions (DMRs) were identified using the DMRcate R package with the default settings [26]. Briefly, this identifies regions of two CpG sites or greater, with an average β value difference of >0.2. The lambda value (maximum distance allowed between neighbouring CpG sites) was set at 1000 bp. Identification of candidate synthetic lethal genes was carried out as previously described [10]. Briefly, methylation was compared in the subgroup of interest (subgroup-2) to all other subgroups grouped together and DMRs identified with the regions of biggest change >0.3 (i.e., a region within the DMR containing at least two CpG sites with an average difference between the two groups of >0.3 in beta value). To allow for the identification of synthetic lethal candidates, as opposed to tumour-suppressor candidates, the focus was specifically on those DMRs with reduced methylation levels in subgroup-2. Regions meeting these criteria were then examined in individual comparisons (i.e., subgroup-2 vs. subgroup-1, subgroup-2 vs. subgroup-3, etc.) to identify those in which reduced methylation was specific for subgroup-2 (i.e., persisted in all the individual comparisons). DMRs that were still retained were then mapped to the genome to identify the nearest gene (DMRs > 20 kb from the nearest gene transcriptional start site were deemed not gene associated and not studied further). Expression was assessed at the identified nearest genes and the candidate retained in expression was higher in subgroup-2 than all other subgroups.

## Figures and Tables

**Figure 1 ijms-24-06387-f001:**
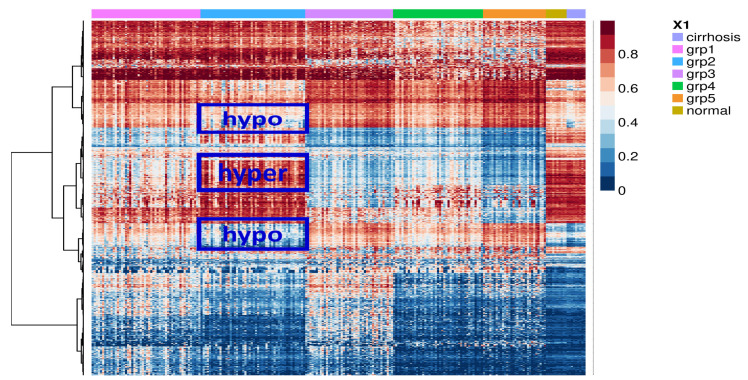
Heat map of DNA methylation at CpG sites in HCC, normal hepatocytes, and cirrhosis samples. Genome-wide DNA methylation data from 221 primary HCC samples [13] was used for NMF analysis to allow for the identification of potential subgroups. This analysis supported the existence of five putative methylation-based HCC subgroups. The figure shows clustering based upon the 10,000 most differentially methylated probes. Regions of methylation difference that are specific for subgroup-2 (versus the other four HCC clusters) are highlighted in blue boxes. G: Subgroup (1 to 5). N: normal hepatocytes. C: Cirrhosis.

**Figure 2 ijms-24-06387-f002:**
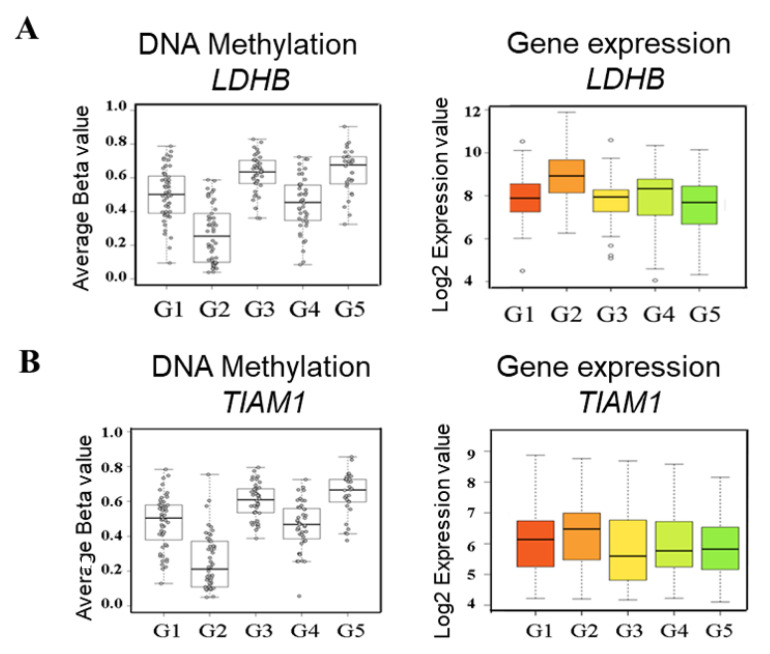
DNA methylation and gene expression of *LDHB* (**A**) and *TIAM1* (**B**) genes in primary HCC patients. G1–G5 refers to HCC subgroups 1–5. Box plots for DNA methylation represent average of beta values for CpG sites identified as the region of maximal difference within DMR corresponding to the indicated gene. Beta value 1.0 is equivalent to 100% methylation, and 0.0 is equivalent to 0% methylation. Box plots for gene expression represent log2 transcript levels of the indicated gene in HCC subgroups.

**Figure 3 ijms-24-06387-f003:**
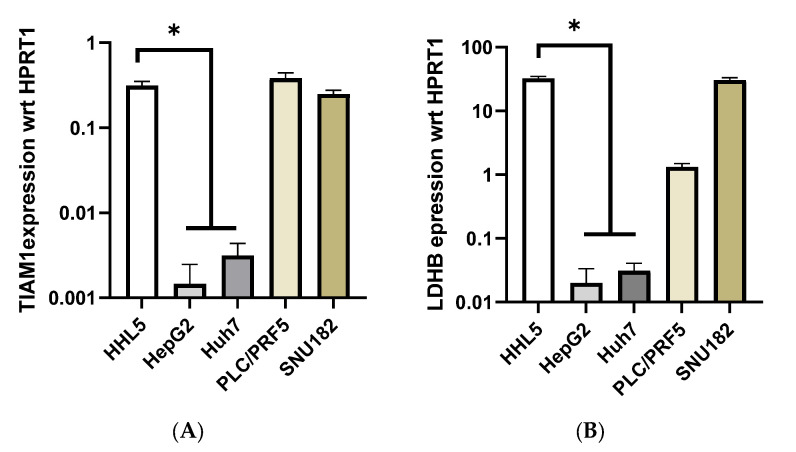
Relative *TIAM1* and *LDHB* gene expression normalized with internal hypoxanthine phosphoribosyl transferase (HPRT1). (**A**) The relative *TIAM1* expression compared with *HPRT1* and (**B**) the relative *LDHB* expression compared with *HPRT1*. Data are calculated as 2^−ΔCt^, and values are shown as mean ± SEM (*n* = 4). ΔCt is the Ct value of targeted *TIAM1* or *LDHB* gene minus Ct value of the housekeeping gene (*HPRT1*). * *p* < 0.0001, for one-way ANOVA with multiple comparisons using HHL5 as the base sample.

**Figure 4 ijms-24-06387-f004:**
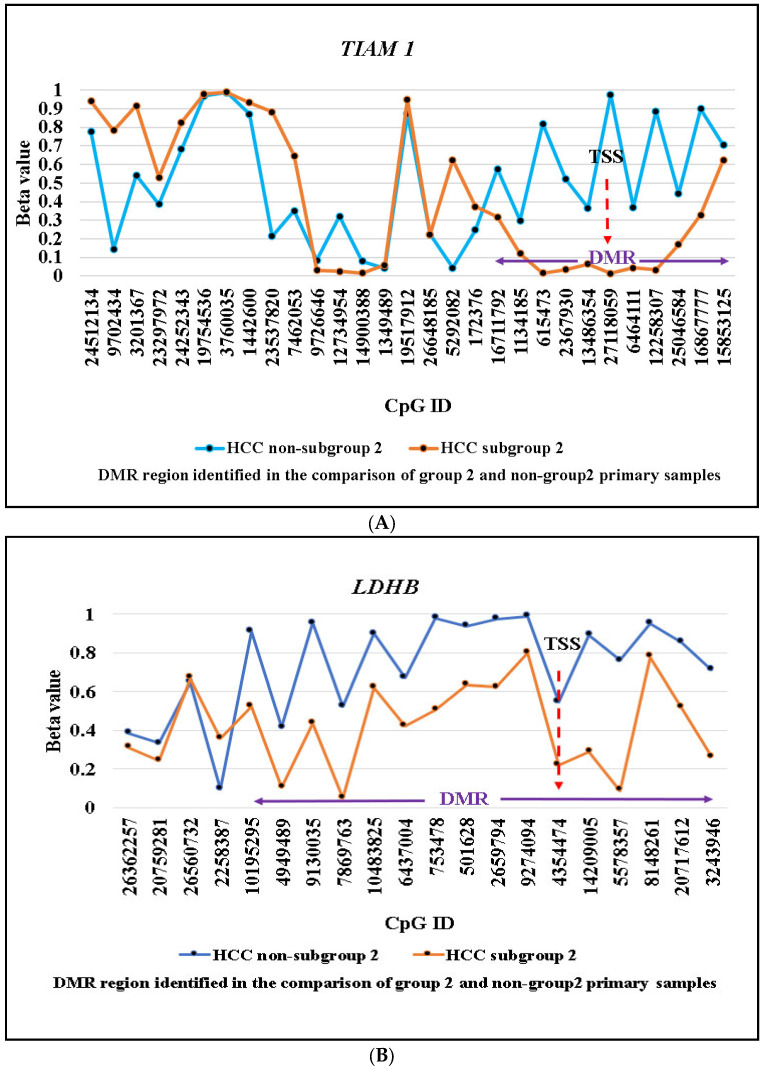
DNA methylation levels at DMR and non-DMR of two candidate SLGs–*TIAM1* (**A**) and *LDHB* (**B**) using Infinium MethylationEPIC array. ID: The identification number refers to the location of individual CpG which can be used as a reference database (CpG loci lack of nomenclature). TSS: transcription start site, indicated by red arrow.

**Figure 5 ijms-24-06387-f005:**
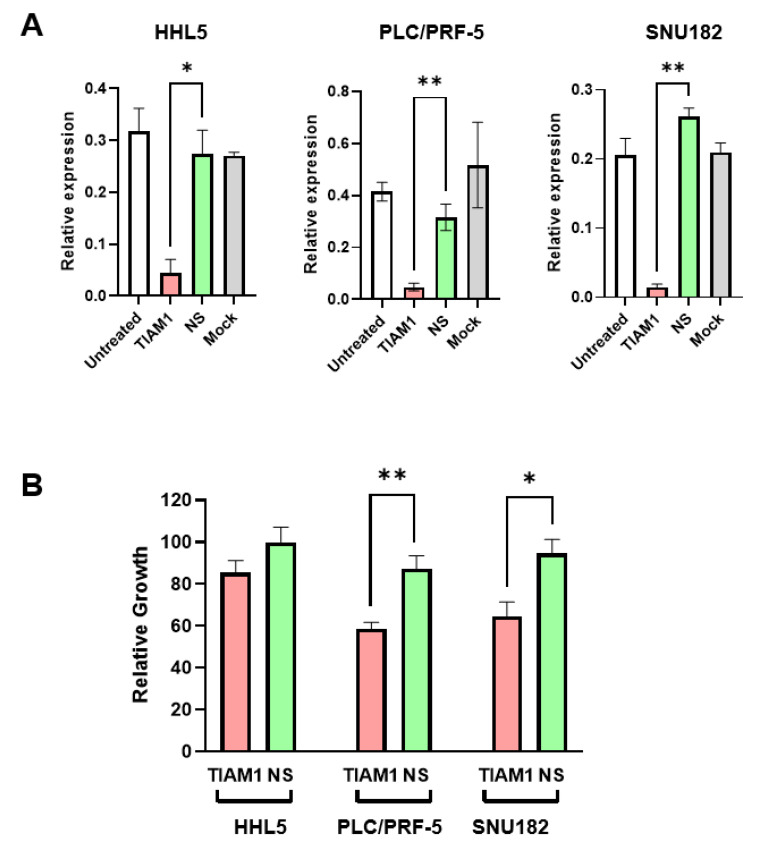
TIAM1 knockdown results in reduced growth in HCC subgroup-2 cell lines. (**A**) RT-qPCR results of cells 48 h post-transfection with indicated siRNAs (TIAM1 or NS (non-silencing)) or controls (Unt = untreated, Mock = only transfection reagent treatment) in HHL5, PLC/PRF-5, and SNU182. Data were calculated as 2^−∆Ct^ relative to the housekeeping control HPRT. Values shown are mean ± SE. * *p* < 0.05, ** *p* < 0.005, Student’s T-test assuming equal variance, *n* = 2–4 (**B**) Cell growth in HCC cell lines following treatment with TIAM1-specific or control non-silencing (NS) siRNA as determined by MTT assay. Graph represents relative growth in indicated conditions compared to growth of respective cell line under mock treated conditions, 6 days after the transfection. * *p* < 0.05, ** *p* < 0.005, Student’s T-test assuming equal variance, *n* = 5.

**Table 1 ijms-24-06387-t001:** The cytotoxicity of NSC23766 on HCC and non-HCC-subgroup determined by MTT assay.

Cell Lines	IC50 (µM)	HCC Subgroup-2?	Average IC50 (µM)	*p*-Value *
PLC/PRF-5	24.8 ± 0.9	Yes	26.3	-
SNU182	27.8 ± 5.2	Yes		
HepG2	57.1 ± 9.6	No	56.2	0.007
Huh-7	49.9 ± 0.1	No		
HHL5	61.6 ± 5.9	Non-HCC		

* Student’s *t*-test assuming equal variance. Table contains mean ± SEM of IC50, *n* = 3.

## Data Availability

The DNA methylation data generated in the study can be accessed at the Gene Expression Omnibus database using accession number GSE228256.

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
