# Peer review of "Identification of TIAM1 as a Potential Synthetic-Lethal-like Gene in a Defined Subset of Hepatocellular Carcinoma"

_ijms, 2023, doi:10.3390/ijms24076387_

Round 1

Reviewer 1 Report

The synthetic lethality phenomenon is very “hot” topic in context of anticancer therapies. Identification of new targets in various cancer types gives new hope for the patients. That is why I see much potential in the work of Permtermsin et al. However, there are some concerns that must be addressed before paper could be published:

11.       Figures:

Firstly, they should be uploaded in higher resolution. Secondly, the bars and error whiskers should not be both black as this make graphs illegible. Thirdly, abbreviation used in axis description are not always clear (e.g. Figure 5a – what is Unt, NS, Mock and TIAM1? In Figure 5b mock treated conditions are marked as NS). Finally, p values are not always shown in the graphs (e.g. Figures 3 and 5).

22.       Why expression level of the two studied genes were made in quadruplicate, when you examine 6 cell lines, while replication varies (2-4 times), when you examine effect of siRNA?

33.       Table 1 lacks description of shown errors and number of replications. Moreover, full results from this experiment should be added, if necessary in supplementary data. Furthermore, the results of MTT should be confirmed using flow cytometry and, ideally, type of cell death could be assessed.

44.       As authors stated, the inclusion of CRISPR technique would strengthen their results.

55.       Why RT-qPCR confirmation of knockdown was performed 48 hrs. after the transfection, while MTT assay after 6 days?

66.       Primers for RT-qPCR were designed by you team? If so, describe to what extent the qPCR-analysis was carried through according to the MIQUE-guide lines (https://doi.org/10.1373/clinchem.2008.112797)?

Author Response

The synthetic lethality phenomenon is very “hot” topic in context of anticancer therapies. Identification of new targets in various cancer types gives new hope for the patients. That is why I see much potential in the work of Permtermsin et al. However, there are some concerns that must be addressed before paper could be published:

Thank you for your comments and appreciating the manuscript. Please see our point-by-point response below:

  1. Figures:

Firstly, they should be uploaded in higher resolution. Secondly, the bars and error whiskers should not be both black as this make graphs illegible. Thirdly, abbreviation used in axis description are not always clear (e.g. Figure 5a – what is Unt, NS, Mock and TIAM1? In Figure 5b mock treated conditions are marked as NS). Finally, p values are not always shown in the graphs (e.g. Figures 3 and 5).

Thank you for your comment. We have made amendments in the figures to address all the issues raised and have included details of the abbreviations in the figure legend in the revised manuscript. We have also carried out statistical analysis in Fig 3 but instead of writing the p values in the graph for Fig 3 and 5, the significant differences are annotated on the graph using asterisk marks and they are defined in the figure legends.

  1. Why expression level of the two studied genes were made in quadruplicate, when you examine 6 cell lines, while replication varies (2-4 times), when you examine effect of siRNA?

Thank you for your comment. We had carried out the siRNA experiment first in PLC/PRF-5 cell line (n=4) and then repeated it in HHL5 and SNU182 cell lines. Since the reduction in expression for all cell lines was very clear and consistent, we repeated the experiment in the additional cell lines only twice.

  1. Table 1 lacks description of shown errors and number of replications. Moreover, full results from this experiment should be added, if necessary in supplementary data. Furthermore, the results of MTT should be confirmed using flow cytometry and, ideally, type of cell death could be assessed.

Thank you for your comment. We have added descriptions of error and repeats in the table legend in the revised manuscript. In addition, we have added the growth inhibition curve as supplementary fig S2. To confirm MTT assay results, we have carried out Caspase 3/7 assay to assess apoptosis using Caspase 3/7 glow kit (Promega). NSC23766 induced apoptosis only in SNU182 cells. The results were confirmed by an independent fluorescence-based cell toxicity assay (CellTox cytotoxicity assay, Promega). The data is included as supplementary Fig S3 in the revised manuscript.

  1. As authors stated, the inclusion of CRISPR technique would strengthen their results.

Thank you for your comment. While we agree that this potentially be helpful, as already stated in the discussion, this would be challenging as the standard CRISPR approach, using selection for stable inactivation would not be possible for functional synthetic lethal genes. There are some recently reported inducible approaches that may be utilisable for functional synthetic lethal genes, but assessing such approaches was not possible within the scope of this current project.

  1. Why RT-qPCR confirmation of knockdown was performed 48 hrs. after the transfection, while MTT assay after 6 days?

Thank you for your comment. The cells were grown for 6 days while doing the MTT assay to clearly assess the influence of knockdown on cell growth. The doubling time of these cell lines in 36-48 hr, thus this period of time is required to allow for 2-3 population doublings after knockdown (determined at 48 hr) and so allow any impact on cell growth to be readily detectable.

  1. Primers for RT-qPCR were designed by you team? If so, describe to what extent the qPCR-analysis was carried through according to the MIQUE-guide lines (https://doi.org/10.1373/clinchem.2008.112797)?

Thank you for your comment. We have designed the primers in-house using NCBI primer blast software. Details of primer optimisation and analysis according to MIQUE guidelines is added to the manuscript and supplementary Table S2.

Reviewer 2 Report

In this manuscript, the authors searched for HCC subgroups that show a distinct methylation pattern. Overall, this is an interesting study, and the data are clear. However, there are several concerns as follows:

Synthetic lethality is not an appropriate term for this study. Please refer to Nature 2022;604:749–756. The authors should use "addicted oncogenes" or "oncogene addiction" or other terms. Although the authors stated that SL genes should be expressed and methylated similarly between a specific tumor and normal cells, they are not necessary for SL genes.

Figure 1: Although G2 shows a distinct methylation pattern from other groups, it is quite similar to those in normal hepatocytes and cirrhotic livers. In addition, those in normal hepatocytes and cirrhotic livers are also similar each other. These point should be discussed. Clustering tree for subgroups should be shown.

The clustered groups (G1~G5) should be analyzed more, for example, clinical relevance (stages and prognosis), transcriptome/proteome, mutation signature, and so on.

More detailed methods of analyses, in particular, how TIAM1 and LDHB genes were identified, should be described.

Author Response

In this manuscript, the authors searched for HCC subgroups that show a distinct methylation pattern. Overall, this is an interesting study, and the data are clear. However, there are several concerns as follows:

Thank you for your comments and appreciating the manuscript. Please see our point-by-point response below:

Synthetic lethality is not an appropriate term for this study. Please refer to Nature 2022;604:749–756. The authors should use "addicted oncogenes" or "oncogene addiction" or other terms. Although the authors stated that SL genes should be expressed and methylated similarly between a specific tumor and normal cells, they are not necessary for SL genes.

Thank you for your comment. We agree that the statement ‘SL genes should be expressed and methylated similarly between a specific tumor and normal cells’ is not true for all SL genes. However, similar methylation patterns between normal cells and SL genes is a key feature of the SL genes identified by our novel bioinformatic approach. This is now clarified in the manuscript (page 2 last line to page 3 line 4).  We agree with the reviewer that the use of the term synthetic lethal genes may be confusing. However, as no oncogenic role has been demonstrated for TIAM1 in this case, oncogene addiction would also not be appropriate. We have therefore referred to the genes as ‘subtype specific vulnerability (SSV) genes’ or ‘SL-like’ genes to clarify that the identified genes are predicted to be specifically required only in a particular disease subtype. We have changed the terminology throughout the manuscript and updated the title accordingly.

Figure 1: Although G2 shows a distinct methylation pattern from other groups, it is quite similar to those in normal hepatocytes and cirrhotic livers. In addition, those in normal hepatocytes and cirrhotic livers are also similar each other. These point should be discussed. Clustering tree for subgroups should be shown.

Thank you for your comment. From our previous analysis of other cancer types [ref 10, Schwalbe et al, Oncogene, 2021], SL-like genes identified by the bioinformatic approach we have developed are predicted to exhibit methylation patterns similar to normal cells and, consistent with this, the regions in subgroup-2 that are differential methylated in comparison to the other four subgroups, typically strongly resemble normal liver tissue. We have added this discussion to the revised manuscript. We haven’t added a clustering tree in Fig 1 as the groups weren’t derived by clustering, but were instead based on NMF.

The clustered groups (G1~G5) should be analyzed more, for example, clinical relevance (stages and prognosis), transcriptome/proteome, mutation signature, and so on.

Thank you for your comment. We have carried out UMAP clustering of these groups using GEO2R online tool, but the groups do not show any distinguished clusters taking whole transcriptome into consideration. Hence, we have not carried out any further transcriptome analysis. The clustering map for the samples is included as supplementary Figure S1 in the revised manuscript. Unfortunately, no clinical data or proteome data is available for these samples in a public domain.

More detailed methods of analyses, in particular, how TIAM1 and LDHB genes were identified, should be described.

Thank you for your comment. We have added further details about the method in the revised manuscript (page 13 line 13 – line 25).

Round 2

Reviewer 1 Report

I think that authors made sufficient changes in a manuscript.

Reviewer 2 Report

The authors have adequately addressed my concerns. I have no further comments.